# Promising Application, Efficient Production, and Genetic Basis of Mannosylerythritol Lipids

**DOI:** 10.3390/biom14050557

**Published:** 2024-05-05

**Authors:** Dun Liu, Guanglei Liu, Shiping Liu

**Affiliations:** 1College of Marine Life Science, Ocean University of China, Qingdao 266003, China; liudun@stu.ouc.edu.cn; 2State Key Laboratory of Resource Insects, Southwest University, Beibei, Chongqing 400716, China

**Keywords:** mannosylerythritol lipids, fungi, biological activity, efficient production, biosynthetic pathways, gene clusters

## Abstract

Mannosylerythritol lipids (MELs) are a class of glycolipids that have been receiving increasing attention in recent years due to their diverse biological activities. MELs are produced by certain fungi and display a range of bioactivities, making them attractive candidates for various applications in medicine, agriculture, and biotechnology. Despite their remarkable qualities, industrial-scale production of MELs remains a challenge for fungal strains. Excellent fungal strains and fermentation processes are essential for the efficient production of MELs, so efforts have been made to improve the fermentation yield by screening high-yielding strains, optimizing fermentation conditions, and improving product purification processes. The availability of the genome sequence is pivotal for elucidating the genetic basis of fungal MEL biosynthesis. This review aims to shed light on the applications of MELs and provide insights into the genetic basis for efficient MEL production. Additionally, this review offers new perspectives on optimizing MEL production, contributing to the advancement of sustainable biosurfactant technologies.

## 1. Introduction

Biosurfactants are amphiphilic compounds synthesized by plants, bacteria, and filamentous fungi as secondary metabolites, are becoming increasingly popular in industrial production, and have the potential to replace fossil-driven surfactants [1,2,3,4,5,6]. According to their chemical properties, biosurfactants can be divided into five main categories: lipopeptides, fatty acids, biopolymers, phospholipids, and glycolipids [7,8]. Mannosylerythritol lipids (MELs) are the most promising natural glycolipid biosurfactants due to their unique properties, including non-toxicity, easy biodegradability, environmental compatibility, and pharmacological activities [9,10,11,12,13,14]. MELs are chemically composed of mannose, erythritol residue, and two fatty acids chains. Their chemical structures vary based on the number and position of the acetyl group on mannose and erythritol, as well as the length and saturation degree of fatty acid chains [10,15,16]. According to the degree of acetylation at C_4_ and C_6_ positions on mannose, MELs are classified into four typical homologues including MEL-A (diacetylated at mannosyl C-4′ and C-6′), MEL-B (monoacetylated at mannosyl C-6′), MEL-C (monoacetylated at mannosyl C-4′), and MEL-D (deacetylated) (Figure 1) [10,12,15,16,17]. The unique structures and features of different MELs expand their applications in various industries such as nanomaterials, bioscience, macromolecular chemistry, cosmetics, and medicines.

MELs were first discovered in 1955 as oily compounds in the cultivation suspension of *Ustilago maydis* PRL-627 and PRL 317 [18], and were later identified as a mixture of partially acylated derivatives of 4-O-β-D-mannopyranosyl-D-erythritol [19]. MELs are produced from feedstock by various fungi of the Ustilaginaceae family such as dimorphic basidiomycete *U*. *maydis* and basidiomycetous yeast-like fungi belonging to the genus *Pseudozyma* [15,20]. Some of these MEL-producing *Pseudozyma* species, such as *P. antarctica*, *P. aphidis*, *P. parantarctica*, and *P. rugulosa*, have been transferred to the genus *Moesziomyces* [21]. The yeast strains *M. aphidis* B1 and *P. hubeiensis* TS18 collected from brown algae and mangrove sediments are excellent producers of MELs [22]. Since MELs are produced by various fungal species, excellent strains and fermentation process are critical for efficient production. For decades, many efforts have been made to improve the fermentation production of MELs by screening high-yielding strains and optimizing the culture medium constituents.

The final fatty acid composition of MELs is influenced by the substrate added to the production medium [6,23]. The optimal production processes for MELs typically use plant oils as a hydrophobic carbon source, with soybean, rapeseed, and olive oil being frequently utilized substrates [10]. The main challenge for large-scale production of MELs is the high production cost due to low fermentation level and the uneconomical downstream processes [24]. Important strategies to reduce the overall production cost for MELs mainly involve screening for high-production strains, utilizing renewable raw materials, developing efficient and cost-effective downstream processing, and optimizing by-products in the culture medium. This review highlights the biological properties and myriad of applications of MELs, strategies for their efficient production, and the genetic basis for the synthesis of these compounds.

## 2. Diverse Biological Activities Endow MELs with Promising Applications

The wide applications of MELs have been exploited in many fields, including agriculture, food, medicine, pharmaceuticals, nanomaterials, healthcare, and cosmetics, mainly depending on their unique biological activities (Figure 2). 

### 2.1. Interaction of MELs with Proteins

Interaction of surfactants with proteins influences the stability and behavior of proteins, and protein−surfactant interactions have been widely used in the food, cosmetic, and medicine industries, and analytical biochemistry [25,26,27,28]. Glycolipid biosurfactants can participate in diverse cell events including signal transduction, cell adhesion, antigenicity, and cell cycle through protein–carbohydrate and/or protein–protein interactions, so it is of great interest to understand the interaction mode of MELs toward various biological macromolecules associated with cellular recognition events [29]. The functional properties of the four types of MELs vary with their structural diversity, and in particular, MEL-A can easily form giant vesicles or nanostructures that interact with biomolecules due to its excellent surface and interfacial tension-lowering activity [30,31,32,33,34]. Interaction between MEL-A and β-glucosidase determines the color stability of anthocyanin beverages given that the structural conformation and enzyme activity of β-glucosidase are inhibited when MEL-A binds to the residues of β-glucosidase, and MEL-A is a potential additive used to change the enzyme properties in the anthocyanin production process [35] (Figure 2A). Accordingly, MEL-A can be used to maintain or improve the flavor and quality of wine or juice.

Protein–biosurfactant interactions significantly influence food functionality. The effects of MEL-A on the gluten network of frozen dough, bread quality, and microbial spoilage have been investigated [4]. MEL-A promotes the formation of aggregates and strengthens the gluten network by interacting with gluten proteins, resulting in significantly improved rheological properties of frozen dough and reduced frozen water content [36]. Elevated concentrations of MEL-A have been shown to decrease moisture migration and improve water-holding capacity in frozen dough, facilitating increased incorporation of starch granules into the dough matrix and positively impacting the flavor profile of steamed bread [37]. The milk protein β-lactoglobulin (β-lg) dominates the structural and functional properties of whey aggregates in food products, and whey aggregates have been widely applied in food products for emulsification, texture modification, thickening, or foaming [25]. Interactions between biosurfactants and β-lg have attracted much attention due to their potential effects on the interfacial properties of food systems, and in particular, efforts have been made to investigate the binding mode of MEL-A toward β-lg and the influence of MEL-A on the formation process of β-lg aggregates [38]. Fan et al. [31] expounded on the biosurfactant–protein interaction by using the MEL-A produced by *M*. *aphidis* DSM70725 and found that MEL-A promotes the self-assembly of heat-induced β-lg and contributes to the formation of spherical particles of MEL-A-β-lg complexes (Figure 2A). The formation of the MEL-A-β-lg complex is driven by the weak hydrophobic interactions between the hydrophobic chains of MEL-A and the nonpolar groups exposed from the heat-induced β-lg protein and the hydrogen bonding between the mannosyl-D-erythritol group of MEL-A and amino acids [38]. In summary, MEL-A is a favorable surface additive to improve the functional properties of β-lg and has broad application prospects in the storage stage of flour products as well as in the whole bread making industry.

The interaction of MEL-A with the heat-induced soy glycinin (11S) aggregates is responsible for surface behavior changes and helps to enhance the foaming stability and emulsifying property of 11S aggregates upon heat treatment; the dominating driving forces of the molecular interaction are the hydrophobic interactions between the exposed hydrophobic groups of the protein and the fatty acid chain or acetyl group of MEL-A, as well as the hydrogen bonding between MEL-A’s mannosyl-D-erythritol groups and 11S’s amino acids [39]. Together, the interaction of MEL-A with 11S aggregates improves the foaming stability and emulsification properties of 11S, suggesting a variety of potential applications of 11S-MEL-A mixtures as natural food additives and emulsions in the beverage and baking industries.

### 2.2. MELs Induce Cell Differentiation in Mammalian Cells

An increasing number of studies have confirmed that MELs can promote the differentiation of different lines of mammalian cells (Figure 2B). Early in 1997, Isoda et al. found that MEL-A or MEL-B markedly induce human promyelocytic leukemia cell line HL60 to differentiate into granulocytic cells, while sophorose lipid, succinoyl trehalose lipid-l, and succinoyl trehalose lipid-3 induce differentiation into monocytes [40]. MEL-A and MEL-B are only different from other glycolipids in their numbers of acetyl groups, so the numbers of fatty acids are critical to the differentiation direction due to their affinities to the cell membranes. MEL-A or MEL-B can also induce the human myelogenous leukemia cell line K562 and the human basophilic leukemia cell line KU812 to differentiate into granulocyte lineages; the differentiation induction in monocytes and granulocytes is not attributed to surface activities of MELs, but to a specific action on the cell membrane [41,42].

Bioactive effects of MELs on B16 cells are in a time-dependent and dose-dependent manner [43]. Mechanically, MEL-induced differentiation of B16 cells might be attributive to different dominant pathways regulated by related signal cascades at specific concentrations of MEL [44]. Differentiation markers of melanoma cells such as protein kinase Cα (PKCα) functioning in the proliferation and differentiation of cells are stimulated upon treatment with MEL (5 µM) for 48 h, indicating that the differentiation of B16 cells is triggered by MEL through the PKCα signaling pathway [44]. MEL-A exhibits a strong inhibition against B16 cell growth and apoptosis in a time-dependent and dose-dependent manner, but the induced cell cycle arrest happens at the S phase, which is different from a previous study suggesting that cell cycle arrests at the G1/G0 phase [43].

MEL can also induce the differentiation of nerve cells. PC12 cell is a pheochromocytoma cell line established from *Rattus norvegicus*, which is similar to the primary culture of fetal rat’s neurons and differentiates to neuronal cells when treated with nerve growth factor, and as a result, has been proven to be a useful cell model system for the study of numerous problems in neurobiology and neurochemistry [45,46,47]. MEL-A and MEL-B induce neurite initiation of PC12 cells during 24 h treatment and a maximal response of the neurite outgrowth can be observed at 48 h after treatment; MEL-A is similar to nerve growth factors in increasing acetylcholinesterase activity in PC12 cells, but the number of neurites induced from the cell body and the differentiation time of the induced cell body are very different, suggesting that MEL-A and nerve growth factors induce PC12 cell differentiation through different mechanisms [47]. Mechanically, MEL triggers the differentiation of PC12 cells to neuronal cells via an ERK-related signal cascade that is partially overlapping the pathways activated in response to NGF [48].

### 2.3. Antibacterial Properties of MELs

Many pathogenic microorganisms, such as Gram-positive bacteria *Staphylococcus aureus*, *Bacillus cereus*, and *Listeria monocytogenes*, are widely distributed in nature and have been recognized as leading concerns to public health and food safety due to their intrinsic virulence, infectious ability, vigorous survival capacity in extreme environments, and their intrinsic and acquired tolerance to commonly used antimicrobial agents [49,50,51,52]; so, novel reagents with the capacity to inhibit and eliminate these pathogenic organisms through antibacterial and antibiofilm activities are urgently needed in the food, cosmetic, medical, and health industries. Biosurfactants have been recognized as powerful antimicrobials and anti-adhesive agents [53]. The influences of MELs on Gram-positive and Gram-negative bacteria are significantly different; for instance, MELs show no obvious effect on Gram-negative *Escherichia coli* and *Proteus vulgaris* but exhibit bactericidal properties particularly against Gram-positive bacteria *S. aureus* and *Bacillus subtilis* [11,50,54,55,56] (Figure 2C). Both the length of the alkyl chains and the pattern of acetyl groups on the mannose moiety of chemically synthesized homogeneous MELs are important for the antibacterial activity against several Gram-positive bacteria [14,54]. MEL-A exerts an excellent inhibitory effect on different growth stages of *L. monocytogenes* by causing destruction of the cell membrane, leakage of intracellular substances, and cell death; MEL-A treatment dramatically reprograms gene expression in the pathways of the maltodextrin ABC transporter system and other important pathways involved in stress response system [55].

The antibacterial properties of MEL are also exemplified by IL-CS-Nano-MEL, modified chitosan nanoparticles with MEL-A as the emulsifier which exhibit enhanced antibacterial activity against *S. aureus* compared to IL-CS-Nano [57]. Much attention has been paid to the green synthesis of gold nanoparticles (AuNPs) using microbial MELs; the biosynthesized MEL-AuNPs show cytotoxicity against HepG2 cells and significantly inhibit the cell growth of pathogenic Gram-positive and Gram-negative bacteria [58]. Essential oils produced by plants exhibit antifungal activity against *Candida* species [59]. After nanoemulsification with MELs, the antimicrobial activity of essential oils, especially *Thymus vulgaris* and *Lippia sidoides*, is effectively preserved and enhanced; so, MELs have the potential as a favorable emulsifier to examine the antimicrobial activity of these essential oils and the application of MELs is quite promising for food packaging applications [60]. According to a recent study [61], the antibacterial activity of MEL is directly related to the length of the alkyl chain, and chemically synthesized MEL-D with decanoyl groups (C10) 18 possesses the most desirable antibacterial activity against Gram-positive bacteria. In conclusion, traditional methods to eliminate foodborne pathogens through high concentration of chemicals or heat treatments have a serious impact on food quality, and MEL is becoming the most promising natural antibacterial agent in the food industry because of its unique antibacterial properties.

### 2.4. Antioxidant Properties of MELs

Essential oils (EOs) are natural agents extracted from different plant parts such as flowers, leaves, fruits, and stems through steam distillation, hydrodistillation, or solvent extraction, and they contain dozens of volatile, fat-soluble, and strongly odorous compounds [62,63]. These compounds are effective in controlling microorganisms which cause biodeterioration and disease and are also potential alternatives for tick-control technologies [64], but their application is restricted mainly because they are easily oxidized, deteriorated, or rapidly vaporized when exposed to oxygen, light, and heat [65]. In recent years, an encapsulation technology comprising the inclusion of surfactants has provided an effective approach to effectively prevent the loss of volatile ingredients, maintain EO stabilization, and improve EO solubility and dispersion [66,67]. By assaying the 1,1-diphenyl-2-picrylhydrazine (DPPH) free radical scavenging activity, Takahashi et al. [68] demonstrated that MEL-C produced from soybean oil by *P*. *hubeiensis* exhibits antioxidant and protective effects in NB1RGB cells (human skin fibroblast) under H_2_O_2_-induced oxidative stress (Figure 2D); further, the amphiphilic structure of MELs can promote the membrane permeability, thus favoring their biological activities between the surfactant and the EOs.

Emulsions prepared with MELs increase antioxidant capacity of three EOs, *Thymus vulgaris* (7.33%), *Lippia sidoides* (13.71%), and *Cymbopogon citratus* (3.15%) [60]. The coating materials MEL-A and fungal chitosan endue liposomes with increased antioxidant capacity, and so the fungal chitosan-coated liposomes modified with MEL-A are a promising delivery system with enhanced antioxidant effects [69]. Anthocyanins are bioactive compounds known for their scavenging properties against reactive oxygen radicals [70]. The vesicles prepared from the mixed solution of MEL-A and L-α-phosphatidylcholine can well encapsulate anthocyanin and enhance its antioxidant capacity during intestinal digestion, which is beneficial for the anthocyanin delivery system [71]. Together, MELs can effectively improve the antibacterial and antioxidant capacity of emulsions (Figure 2D), and are thus quite promising for food packaging applications and oxidative damage.

### 2.5. Skin and Hair Care Properties of MELs

The epidermis serves as a protective barrier against pathogens, irritants, and ultraviolet radiation, while also regulating the loss of water and solutes to maintain cellular homeostasis. The integrity of the skin barrier is determined by a specialized, stratified structural protein complex, the efficacy of which is contingent upon the interplay among transglutaminase-crosslinked proteins, including filaggrin (FLG), loricrin (LOR), and transglutaminase-1 (TGM1) [72,73,74,75]. Several studies have shown that deficiencies in FLG, LOR, and TGM1 result in skin disorders like dryness, flaking, lipid changes, and allergies [76,77,78,79,80]. Aquaporins (AQPs) are a class of integral membrane proteins that modulate water movement across the plasma membrane, thus contributing to the regulation of water homeostasis in the epidermis [81,82,83]. Reduced AQP3 in the skin leads to dryness, less elasticity, lower glycerol levels, and poor wound healing [84,85]. The expression levels of AQP3 are significantly lower in non-sun-exposed human skin of individuals aged over 60 years compared to those under 45 years, showing that AQP3 potentially plays a role in the endogenous aging mechanism of non-sun-exposed human skin [86]. The decline in AQP3 expression in human skin keratinocytes can also be induced by detrimental external factors, including ultraviolet (UV) irradiation and reactive oxygen radicals [87]. In 2009, Morita et al. conducted a study on the skin care properties of MEL-A using a three-dimensional cultured human skin model, which demonstrated that MEL-A significantly increases cell viability and exhibits ceramide-like moisturizing effects on skin cells [88]. Three years later, they examined the impacts of various MELs on damaged skin cells using a three-dimensional cultured human skin model and evaluated the effects of MELs on the water retention properties of the skin using an in vivo human study, and the results showed that both MEL-A and MEL-B not only have a strong recovery effect on the damaged skin cells but also exhibit a high moisturizing action on the skin by assisting the barrier function of the skin [89] (Figure 2E). In 2019, Bae et al. found that the skin-moisturizing effect of MELs may be mediated by regulating the expression of AQP3 [90] (Figure 2E). In 2022, Jing et al. conducted a study on the protective effects of three compounds (MEL-A, MEL-B, and MEL-C) on skin damage, and the results showed that MEL-B effectively protects human immortalized keratinocytes (HaCaT cells) from UVB-induced damage by upregulating the contents of the expression of LOR, FLG, and TGM1 (Figure 2E), highlighting its promise as a beneficial ingredient for use in skin care products [91]. In conclusion, due to their excellent moisturizing effect, MELs can be added to a variety of skin care products, such as face cream, lotion, and essence liquid, to improve the moisturizing performance of the product.

In addition to unique moisturizing effects on human skin, MELs also exhibit hair care properties against environmental stresses such as UV radiation, pollution, and heat damage [92]. The dermal papilla cells induce follicle formation and hair growth through trans-differentiation of an adult epidermis [93], so activation of the papilla cells is crucial for the development of a new hair growth integrant. Morita et al. [94] demonstrated that MEL-A produced from soybean oil significantly increases the viability of both the fibroblast cells and the papilla cells but MEL-A produced from olive oil exerts no effect on cell activation. Further, they investigated the hair care properties of MELs using damaged hair and revealed that the cracks of damaged hair are repaired and the tensile strength of the damaged hair is increased after treatment with MEL-A or MEL-B, suggesting that MEL-A and MEL-B emerge as promising hair care agents, capable of not only repairing damaged hair but also conferring smoothness and flexibility on the hair [95]. Collectively, MELs are known for their excellent emollient and moisturizing properties, which make them excellent candidates for use in hair care products.

### 2.6. Depigmentary Properties of MELs

Melanocytes are melanin-producing cells present in a variety of tissues in the body. The melanogenesis is a complex process that can be affected by a number of intrinsic and extrinsic stimulatory factors through different intracellular signaling pathways [96,97,98]. In mammals, a multienzyme complex composed of melanocyte-specific gene products coordinates the tightly regulated melanogenesis pathway, allowing for the conversion of tyrosine first to 1-3, 4-dihydroxyphenylalanine (DOPA), then to dopaquinone, dopachrome, and finally to melanin [99]. Excessive production of melanin in melanocytes causes hyperpigmentation, such as freckles and lentigo, so effective depigmentary ingredients are expected to act selectively on hyperactivated melanocytes, without short- or long-term side effects, and to induce permanent removal of undesired pigment [100]. Yeast glycolipid biosurfactants have been proven to moisturize dry skin, repair damaged hair, activate fibroblasts and papilla cells, and play antioxidant and protective roles in skin cells, thus becoming one of the most promising alternatives in cosmetics [101]. 

Recently, researchers have found that MELs also have depigmentation properties, which have potential application value in skin whitening, freckle removal, and other aspects. In 2019, the potential depigmentation effect of MELs was tested in primary normal human melanocytes (NHMs), α-melanocyte-stimulating hormone (MSH)-stimulated B16 cells (mouse melanoma cells), and human skin equivalent cells (MelanoDerm), and it was revealed that MELs significantly inhibit melanin synthesis in NHMs and α-MSH-stimulated B16 cells and exert clear whitening effects on human melanocytes and a 3D human skin equivalent [102]. Further, to investigate the molecular mechanism of MELs against melanin production, the inhibitory effect on tyrosinase activity was tested and the expression levels of melanogenic enzymes (tyrosinase, Tyrp-1, and Tyrp-2) were examined in MEL-treated NHMs, and the results suggest that MELs significantly suppress melanin-producing enzymes through inhibition of the ERK/CREB/MiTF signaling pathway [102] (Figure 2F). Based on the decolorization effect of MEL, researchers have been exploring its applications in cosmetics, pharmaceuticals, and other fields [10,17,92]. In the cosmetics industry, MEL can be added as a whitening ingredient to skin care products [102], helping to improve skin color and make the skin fairer and more radiant. In the pharmaceutical field, MEL can be used to treat skin diseases caused by melanin deposition, such as melanosis and freckles [103]. In addition, MEL can also be used to treat some diseases related to melanogenesis, such as melanoma [31,104]. However, research on the decolorization effect of MEL is still in its early stages, and there are still many problems that need to be solved urgently. For example, is the decolorization effect of MEL applicable to all populations? How safe and effective is its long-term use? The answers to these questions still require further research and exploration.

## 3. Strategies for Efficient Production of MELs

The chemical properties of biosurfactants and their production costs are mainly determined by the selection of production microorganisms, substrate types, and purification strategies [105]. High-yielding strains, optimized fermentation conditions, and the use of cheaper substrates are essential to reduce MEL production costs and expand industrial production [106].

### 3.1. Microorganisms Capable of Efficiently Producing MELs

Due to its distinctive biological properties, MEL has garnered significant interest for its potential applications in various sectors including medicine, cosmetics, and food. The exploration of microorganisms capable of producing MEL efficiently has emerged as a prominent area of research. Numerous filamentous fungi capable of MEL production have been successfully isolated and characterized (Table 1). Smut fungi *U. maydis* and *Schizonella melanogramma* are the first and second microorganisms identified as MEL producers, respectively [18,107]. The organisms that synthesize MELs are mainly found in the *Ustilaginaceae*, a class of basidiomycetes including plant-pathogenic ustilago and non-pathogenic yeasts [108]. Different types of microorganisms show different synthetic efficiency in the synthesis of MELs [108,109,110,111,112,113,114,115] (Figure 3; Table 1). Strains of *M. aphidis* DSM 70725 and DSM 14930 are able to produce high yields of MELs from different vegetable oils [116,117]. A yeast strain of *Candida antarctica* T-34 has long been used for MEL production with soybean oil as the substrate in the culture broth [118,119]. In 2022, three MEL compounds (MEL-A, MEL-B, and MEL-C) were extracted from *C. antarctica* cultures containing fermented olive oil and then purified using silica gel-based column chromatography and semipreparative HPLC [91]; in the same year, a metabolic analysis was conducted on *M. antarcticus* T-34 cultured in olive oil to investigate the crucial pathways involved in oil absorption and MEL biosynthesis [120]. In these cultures, olive oil is enzymatically digested into fatty acids and glycerol through the action of lipase secreted by *M. antarcticus* T-34 and subsequently absorbed by the cells. More recently, a mangrove yeast strain *M. aphidis* XM01 identified from mangrove plants was employed for efficient extracellular MEL production through a two-stage fed-batch bioprocess [121]. In this two-stage fed-batch fermentation, the final MEL titer per 10 L system reached 113.6 ± 3.1 g/L within 8 days, with prominent productivity and a yield of 14.2 g·L^−1^·day^−1^ and 94.6 g/g_(glucose and soybean oil)_; so, the *M. aphidis* XM01 strain is supposed to be an alternative microbial cell factory for industrial production of MELs from soybean oil [121]. Structural composition analysis by using GC/MS showed that the MELs produced by the strain XM01 have good physicochemical stability, antibacterial activity, and encapsulation and release behavior, and are supposed to have broad application prospects in the pharmaceutical and cosmetic fields [121].

Besides the conventional MELs, an increasing number of other MEL derivatives have been reported as well. Fukuoka et al. reported the production of monoacylated MELs produced from glucose by *M. antarctica* [123]. Morita et al. obtained a significant yield of triacylated MELs (22.7 g L^−1^) from the fermentation products of *M. parantarctica* JCM 11752 (T) by increasing the amount of oil substrate to 20% (*v*/*v*) and the fermentation temperature to 34 °C [124]. MELs were produced in a 22 L bioreactor through batch and fed-batch fermentation of rapeseed oil with one high-level MEL-producing yeast, *M. aphidis* MUCL 27852 [125]. During the fermentation process, this strain not only produced a high level of four conventional MEL structures, namely, MEL-A, MEL-B, MEL-C, and MEL-D, but also secreted additional unknown hydrophobic glycolipids which were separated by flash chromatography and identified as triacylated MELs by high-performance liquid chromatography tandem mass spectrometry (HPLC–MS/MS) [125]. Actually, these apolar products were labeled as unknown compounds on thin-layer chromatography in an earlier work [126]. With the optimization of fermentation yield, *P. aphids* could become an interesting novel producer of triacylated MELs, introducing more structural variety amongst biosurfactants and expanding the availability and applicability of biosurfactants. In conclusion, researchers have identified a growing number of microorganisms with the ability to efficiently produce MELs or their derivatives and have optimized their fermentation conditions (Table 1). This development not only establishes a basis for the industrial-scale production of MELs, but also offers valuable insights for the microbial fermentation production of other bioactive compounds.

**Table 1 biomolecules-14-00557-t001:** Microorganisms show different synthetic efficiency in the synthesis of MELs.

MEL Producer	T (°C)	pH	Main Product	Nitrogen Source	Carbon Source	Culture Vessel	MEL Yield (g/L)	Culture Time	Ref.
*Pseudozyma antarctica* T-34	30	ND	MEL-A	NaNO_3_	soybean oil	flask	34	7 days	[119]
*Pseudozyma antarctica* CBS 5955	30	ND	MEL-A	NaNO_3_	soybean oil	flask	8.9	7 days	[119]
*Pseudozyma antarctica* CBS 6678	30	ND	MEL-A	NaNO_3_	soybean oil	flask	16.7	7 days	[119]
*Pseudozyma antarctica* CBS 6821	30	ND	MEL-A	NaNO_3_	soybean oil	flask	27.1	7 days	[119]
*Ustilago maydis* DSM 4500	30	2	MEL-A	(NH_4_)_2_SO_4_ or urea	sunflower oil	flask	30	6 days	[127]
*Pseudozyma* *antarctica* ATCC 20509	30	ND	MEL	NaNO_3_	soybean oil	flask	45.5	6 days	[128]
*Pseudozyma antarctica* ATCC 20509	30	ND	MEL	NaNO_3_	glucose	flask	1.1	6 days	[128]
*Pseudozyma antarctica* T-34	30	5.7	MEL-A	NaNO_3_	soybean oil and n-octadecane	flask	140	4 weeks	[129]
*Moesziomyces aphidis* DSM 14930	27	6	MEL-A	NaNO_3_	soybean oil	flask	90	8 days	[130]
*Pseudozyma aphidis* DSM 70725	30	6	MEL-A	NaNO_3_	soybean oil	flask	40	10 days	[117]
*Pseudozyma aphidis* DSM 70725	27	6.2	MEL	NaNO_3_	soybean oil and glucose	flask	70	8 days	[116]
*Pseudozyma aphidis* DSM 14930	27	6.2	MEL	NaNO_3_	soybean oil	flask	90	8 days	[116]
*Pseudozyma aphidis* DSM 14930	27	6.2	MEL	NaNO_3_	soybean oil and glucose	flask	165	8 days	[116]
*Pseudozyma rugulosa* NBRC 10877	25	6	MEL-A	NaNO_3_	soybean oil	flask	142	4 weeks	[131]
*Candida* sp. SY16 (*Candida antarctica* KCTC 7804)	25	6	MEL-A	NaNO_3_	soybean oil	flask	142	7 days	[131]
*Candida antarctica* KCTC 7804	30	4	MEL-SY16	NH_4_NO_3_	glucose and soybean oil	flask	95	8.3 days	[132]
*Pseudozyma tsukubaensis* JCM 10324T	30	6	MEL-B	NaNO_3_	soybean oil	flask	27	7 days	[109]
*Pseudozyma antarctica* JCM 10317T	25	6	MEL-D	NaNO_3_	glucose	flask	1.1	7 days	[123]
*Pseudozyma parantarctica* JCM 11752T	25	6	MEL-D	NaNO_3_	glucose	flask	1.2	7 days	[123]
Pseudozyma antarctica T-34	25	6	MEL-D	NaNO_3_	glucose	flask	1.3	7 days	[123]
*Pseudozyma antarctica* T-34	25	6	MEL-A	NaNO_3_	D-glucose	flask	12	3 weeks	[133]
*Pseudozyma antarctica* T-34	30	ND	MEL-A	NaNO_3_	soybean oil	flask	20	10 days	[110]
*Pseudozyma* sp. KM-160	30	ND	MEL-B	NaNO_3_	soybean oil	flask	25	10 days	[110]
*Pseudozyma* sp. KM-59	30	ND	MEL-C	NaNO_3_	soybean oil	flask	23	10 days	[110]
*Pseudozyma antarctica* JCM 3941	30	6	MEL-A	NaNO_3_	soybean oil	flask	25	7 days	[109]
*Pseudozyma aphidis* JCM 10318T	30	6	MEL-A	NaNO_3_	soybean oil	flask	23	7 days	[109]
*Pseudozyma fusiformata* JCM 3931T	30	6	MEL-A	NaNO_3_	soybean oil	flask	4	7 days	[109]
*Pseudozyma parantarctica* JCM 11752T	30	6	MEL-A	NaNO_3_	soybean oil	flask	30	7 days	[109]
*Pseudozyma rugulosa* JCM 10323T	30	6	MEL-A	NaNO_3_	soybean oil	flask	26	7 days	[109]
*Pseudozyma shanxiensis* CBS 10075	28	6	Ps-GL	NaNO_3_	soybean oil	flask	2.72	4 days	[134]
*Ustilago cynodontis* NBRC 7530	25	6	MEL-C	NaNO_3_	soybean oil	flask	1.4	7 days	[135]
*Ustilago maydis* NBRC 6707 (DSM 4500)	25	6	MEL-A	NaNO_3_	soybean oil	flask	1.9	7 days	[135]
*Pseudozyma graminicola* CBS 10092	30	6	MEL-C	NaNO_3_	soybean oil	flask	10	7 days	[136]
*Pseudozyma siamensis* CBS 9960	25	6	MEL-C	NaNO_3_	safflower oil	flask	18.5	9 days	[137]
*Pseudozyma shanxiensis* CBS 10075	25	6	MEL-C	NaNO_3_	safflower oil	flask	2.7	9 days	[137]
*Pseudozyma antarctica* JCM 10317	25	6	MEL-A	NaNO_3_	olive oil	flask	12:98	7 days	[115]
*Ustilago maydis* NBRC 5346	25	6	MEL-A	NaNO_3_	olive oil	flask	2.62	7 days	[115]
*Ustilago scitaminea* NBRC 32730	25	6	MEL-B	NaNO_3_	olive oil	flask	8.29	7 days	[115]
*Pseudozyma siamensis* CBS 9960	25	6	MEL-C	NaNO_3_	olive oil	flask	1.35	7 days	[115]
*Ustilago scitaminea* NBRC 32730	25	6	MEL-B	NaNO_3_	sucrose	flask	12.8	21 days	[138]
*Pseudozyma tsukubaensis* IE5(JCM16987)	25	6	MEL-B	NaNO_3_	olive oil	flask	73.1	7 days	[112]
*Pseudozyma tsukubaensis* NBRC1940	25	6	MEL-B	NaNO_3_	castor oil or olive oil	flask	22.2	7 days	[113]
*Moesziomyces aphidis* DSM 70725	30	6	MEL-A	NaNO_3_	rapeseed oil	flask	34.3	7 days	[139]
*Moesziomyces aphidis* XM01	28	6	MEL-A	NaNO_3_	soybean oil	flask	113.6	8 days	[121]
*Moesziomyces antarcticus* PYCC 5048^T^	27	6	MEL	NaNO_3_	soybean oil	flask	50	12 days	[140]
*Moesziomyces aphidis* PYCC 5535^T^	27	6	MEL	NaNO_3_	soybean oil	flask	50	12 days	[140]
*Moesziomyces antarcticus* PYCC 5048^T^	27	ND	MEL	NaNO_3_	soybean oil	flask	19.5	14 days	[141]
*Moesziomyces aphidis* PYCC 5535^T^	27	ND	MEL	NaNO_3_	soybean oil	flask	21.8	14 days	[141]
*Pseudozyma aphidis* DSM 70725	30	6.2	MEL	NaNO_3_	soybean oil	flask	61	8 days	[142]

Note: ND, not detailed in the reference.

### 3.2. Technical Feasibility of Using Inexpensive Fermentation Substrates to Produce MELs

The advancement of biotechnology has led to an increasing demand for efficient, environmentally sustainable production methods. As a natural product with important biological activity, MEL has shown a growing market demand. To achieve cost-effective large-scale production of MEL, utilizing inexpensive fermentation substrates has become a favorable option. Microorganisms can use various compounds as carbon sources to support their growth. A notable limitation in large-scale MEL production is the elevated expense of raw materials, which hinders its sustainable commercial implementation. Therefore, the utilization of cost-effective substrates serves to diminish production costs and enhance the viability and commercial potential of MEL production. Lignocellulose is the most abundant renewable carbon resource, and the efficient conversion of lignocellulosic materials into advanced biofuels and other bio-based products allows for sustainable scale production of bio-based products [143,144,145]. The cellulose-rich substrate, pretreated wheat straw, can be converted into MEL by *M. antarctica* PYCC 5048T and *M. aphidis* PYCC 5535T under separate hydrolysis and fermentation and simultaneous saccharification and fermentation processes [146]. In addition, waste cooking oil can be used as the sole carbon source to produce MELs by *M. aphidis* ZJUDM34, which helps to lower the production cost of MELs [147]. In 2022, Nascimento et al. reported the production of β-galactosidase by *M. aphidis* using different sugars, including D-galactose, D-glucose, and D-lactose [148]. Interestingly, D-galactose was found to be the most effective inducer of β-galactosidase. This enzyme production is significant as it enables the breakdown of D-lactose, allowing for the direct production of MEL from D-lactose and cheese whey. In summary, as biotechnology continues to advance and environmental regulations become more stringent, utilizing cost-effective fermentation substrates such as renewable industrial residues and agro-industrial wastes for the production of MEL presents a promising avenue for future production. This approach not only aids in reducing production costs and enhancing market competitiveness of MEL, but also contributes to sustainable resource utilization and environmental conservation.

### 3.3. Optimization of the Culture Medium and Fermentation Conditions for Enhancing the Production of MELS

The culture medium serves as the fundamental substrate for microbial growth and metabolism, with its formulation playing a crucial role in modulating the production of MELs. When seeking to enhance the efficacy of the culture medium, key considerations typically include the selection and optimization of components such as carbon source, nitrogen source, inorganic salts, and growth factors [106]. Sodium nitrate (NaNO_3_) is commonly employed as a nitrogen source for microbial growth in the production of MELs. For example, a recent study demonstrated that the physiological alkaline salts of NaNO_3_ and potassium nitrate (KNO_3_) led to significantly higher MEL titers of approximately 60 g/L, while the physiological acidic salts of ammonium nitrate (NH_4_NO_3_) and ammonium sulfate ((NH_4_)_2_SO_4_) did not result in MEL production; despite this, NaNO_3_ was determined to be the most effective nitrogen source when considering the cost factors, and the highest MEL yield was achieved when the concentration of NaNO_3_ was 2.0 g/L [121] (Table 1). In 2020, Beck et al. investigated the growth of seven *Ustilaginaceae* species with three different liquid culture media containing the same concentrations of glucose and sodium nitrate but different amount of mineral salts, vitamins, or trace elements, and it was shown that high concentrations of vitamins and trace elements are necessary for the cell growth of *Ustilaginaceae* fungi and successive MEL production from rapeseed oil [149]. Then, they performed fermentative production of MEL in a bioreactor with *M. aphidis* using a defined mineral salt medium and developed kinetic model equations for the precise prediction of the process behavior during cell growth and MEL production phase [139]. This model is able to simulate the time course of biomass and substrate concentrations during batch and fed-batch growth, describe the MEL production process in detail, and provide a better understanding of key economic and ecological parameters for MEL production.

Using a medium with specific trace elements can improve fermentation consistency and process control, benefiting the scaling up of MEL production. In a study conducted by Niu et al. [150], MEL-A was derived from the strain *Ceriporia lacerate* CHZJU. The researchers employed a Plackett–Burman design and response surface methodology to optimize the culture nutrient. They developed a fermentation kinetic model for MEL production in this strain, which is beneficial for enhancing the efficiency of the industrial process. Recently, Yang et al. optimized the fermentation process of *M. aphidis* DSM 70725 by adjusting the levels of Fe^2+^ and Fe^3+^ to improve the fermentation efficacy, and by using soybean oil to reduce the foam produced during the fermentation process [142]. These optimized fermentation processes almost double the yield of traditional fermentation processes and are suitable for large-scale fermentation of MEL that uses soybean oil as a defoamer to overcome foaming issues.

In addition to the formulation of the culture medium, the fermentation conditions also play a crucial role in influencing the production yield of MEL. The researchers primarily focused on optimizing parameters such as temperature, pH, agitation rate, and aeration level [106,149]. Temperature is a key factor affecting microbial growth and metabolism. By comparing the fermentation effects at different temperatures, it was found that 27–30 °C is the most suitable temperature (Table 1) [106,121,140,151]. The pH value has a significant impact on the growth of microorganisms and the synthesis of metabolites, and the optimal pH of different strains may vary significantly. In a fermentation study of MEL-SY16 production from *Candida antarctica* KCTC 7804 conducted by Kim et al., the impact of pH regulation on the synthesis of MEL-SY16 was investigated during batch fermentation and the optimal production output of MEL-SY16 was achieved when pH was maintained at 4.0 [132]. The release of fatty acids is essential for the biosynthesis of MEL, and elevated pH levels may impede the secretion or activity of extracellular lipase necessary for lipid hydrolysis. Consequently, in the study by Beck and Zibek [149], they observed that the yield of MEL was enhanced only when the pH dropped to around 5.5 in mineral medium 2 for *S. graminicola* and *P. tsukubaensis*. In a previous work on the marine yeast *M. aphidis* XM01, the pH value of the fermentation broth was controlled between 6.0 and 7.0 for a higher yield of MEL (Table 1) [121]. Stir speed and ventilation rate can influence the dissolved oxygen levels in the fermentation broth, thereby affecting the growth and metabolism of microorganisms [106]. For example, by adjusting the stirring speed and ventilation rate, it was found that in the pilot-scale fermentation production of MELs from *P. aphidis* DSM 70725, the yield of MEL was higher when the stirring speed was 200 rpm and the ventilation rate was 0.3–0.5 vvm (volume of air added to liquid volume per minute) [142]. In summary, by optimizing the culture medium and fermentation conditions, the yield of MEL from various producers has been increased. However, there are still some issues that require further research, such as the metabolic pathways of microorganisms and the catalytic mechanisms of enzymes. In the future, it is necessary to continue in-depth research on these issues in order to further improve the production efficiency of MEL and promote its development in practical applications.

### 3.4. Optimization of the Downstream Purification Processes

For the production of biosurfactants, the production process is complete only when the product is economically recovered and purified in a proper way, and product recovery from the culture medium is one of the most important parameters for the commercialization of biosurfactant production [151]. The removal of triglycerides and other residual lipid derivatives from microbial fermentation cultures involves the use of a large mixture of organic solvents, which compromises solvent recyclability and increases the final process cost. Inefficient and costly downstream processes of MEL production restrict its large-scale production and application, and so it is highly necessary to develop efficient and sustainable production processes with low residue generation. The use of renewable raw materials can reduce costs and facilitate downstream processing. Faria et al. [141] adopted a co-substrate strategy whereby the production of MEL by *M. antarcticus* and *M. aphidis* was facilitated by the sequential supply of hydrophilic and hydrophobic carbon sources in combination with a novel downstream strategy for MEL purification based on nanofiltration technology. For *M. antarcticus*, this strategy resulted in an MEL yield threefold higher than that obtained with D-glucose as a carbon source, but maintained a low concentration of residual lipids, enabling further downstream processing. By combining the application of diafiltration and cultivation conditions, they also developed a new downstream route to effectively isolate and purify MELs from residual lipids using home-made flat-sheet organic solvent membranes. Desirably, developing membranes with high retention of MEL and/or low retention of residual lipids is critical to optimizing separation and mitigating MEL losses.

Recently, an innovative downstream method for MELs was recently reported by Nascimento et al. [140], which is a unique solution for microbial biosurfactant production with minimal product losses, enabling solvent recycling and potentially reducing costs. According to this report, 90% of the triacylglycerols are separated from the raw MEL mixture in the first stage and the other lipid derivatives (free fatty acids, and mono- and diacylglycerols) are removed through organic solvent nanofiltration, followed by using activated carbon to remove color impurities generated during the fermentation from the MELs, resulting in a pure product with light coloration. Overall, this downstream process provides a unique solution based on a single solvent for the purification of biosurfactants produced from hydrocarbon and lipid-based substrates, avoiding the use of solvent mixtures and enabling solvent recovery and reuse. Future work could include multi-objective optimization techniques to improve biosurfactant production and downstream processing, thereby further reducing the cost of overall MEL production, leading to commercial applications.

## 4. Genetic Basis of MEL Biosynthesis

The biosynthesis of MELs is facilitated through shared biosynthetic pathways (Figure 4), wherein genes responsible for fungal-specific secondary metabolites are typically co-located near telomeres [152]. Following the determination of the chemical structure of these biosurfactants, genomic analyses of MEL producers facilitate the identification of the gene clusters involved in MEL biosynthesis (Figure 5).

### 4.1. Genes Essential for the Production of MELs in U. maydis

The basidiomycetous fungus *U. maydis* has garnered significant attention for its unique biosynthetic capabilities, especially in the production of MELs. The biosynthesis of MELs involves a series of enzymatic reactions and metabolic pathways (Figure 4) [153]. In order to elucidate the genetic basis of fungal glycolipid biosynthesis, many studies have been carried out to determine the candidate genes necessary for the production of this compound. Early in 2005, Hewald et al. cloned two genes *emt1* and *cyp1* required for glycolipid biosynthesis in *U. maydis* [154]. In 2006, they further identified a gene cluster essential for MEL biosynthesis and put forward a biosynthesis pathway for the MELs in *U. maydis* after mutational and biochemical analysis of the MEL biosynthesis cluster [155]. The *U. maydis* MEL cluster consists of five open reading frames, *mat1*, *mac1*, and *mac2* encoding an acetyltransferase (mat1) and two acyltransferases (mac1 and mac2), *emt1* coding glycosyltransferase, and *mmf1* specifying one export protein of the major facilitator family (Figure 5) [155]. In conclusion, a gene cluster is responsible for encoding enzymes involved in the synthesis of MEL, including a glycosyltransferase Emt1, an acetyltransferase Mat1, and two acyltransferases Mac1 and Mac2, which together contribute to the biosynthesis pathway of MEL in *U. maydis*.

A further study conducted by Freitag et al. unveiled a critical aspect of MEL biosynthesis in *U. maydis* that occurs within specialized subcellular compartments called peroxisomes, where the two acyltransferases Mac1 and Mac2, key contributors to MEL synthesis, couple MEL biosynthesis to the peroxisomal β-oxidation pathway (Figure 4) [156]. Peroxisomes are morphologically simple and ubiquitous organelles in virtually all eukaryotes crucial for the primary metabolism of several unusual carbon sources and for the formation of a variety of secondary metabolites [157]. Immunolocalization experiments demonstrated the co-localization of Mac1 and Mac2 within peroxisomes in *U. maydis*, while the glycosyltransferase Emt1 was primarily associated with vacuoles, and the acetyltransferase Mat1 was predominantly observed at the plasma membrane, indicating that different steps of MEL biosynthesis occur in distinct intracellular compartments [156]. Manipulating peroxisome dynamics holds promising prospects for enhancing MEL production, as demonstrated in the case of penicillin-producing strains [156]. In 2021, Becker et al. conducted genetic modifications to produce customized MELs, highlighting the crucial roles of acyl-transferases Mac1 and Mac2 in the process; they combined the genes *Mac1* and *Mac2* from different fungal species *U. maydis* and *U. hordei* to engineer tailor-made novel MEL variants with altered acylation patterns [158]. The identification and characterization of the genes responsible for MEL biosynthesis in *U. maydis* have yielded significant insights into the biosynthetic pathway and established a basis for further investigations aimed at improving MEL production and understanding its various biological functions. Subsequent research endeavors may concentrate on elucidating the complex subcellular localization mechanisms involved in MEL biosynthesis and devising methods to manipulate peroxisomal dynamics to enhance MEL output.

### 4.2. Genetic Basis of MEL Biosynthesis in P. tsukubaensis

Basidiomycetous yeast *P. tsukubaensis* has been revealed to efficiently produce MEL-B when cultured in medium using olive oil as the sole carbon source [109,110,111,112,113,114,159] (Table 1). Subsequently, increasing attention has been paid to the genetic basis of MEL synthesis in these strains, including the genes encoding related enzymes and important regulatory pathways. In 2016, Saika et al. obtained for the first time the gene cluster involved in MEL-B biosynthesis in *P. tsukubaensis*, and demonstrated that PtEMT1p, PtMAC1p, PtMAC2p, and PtMMF1p display significant homology (over 50%) with their counterparts in *U. maydis* and *M. antarctica*, and that *PtEMT1p* encodes the erythritol/mannose transferase catalyzing the sugar conformation of MELs (Figure 5) [160]. The recombinant *P. tsukubaensis* strain 1E5 (JCM16987) with two lipase genes *PaLIPA* and *PaLIPB* from *M. antarctica* T-34 increases MEL-B yield by over 1.9-fold compared to the control strain, suggesting that overexpressing these genes is an effective way to boost production and oil consumption in *P. tsukubaensis* [161]. Indeed, MEL-B production increases twofold when using the translation elongation factor 1 alpha/Tu promoter to overexpress *PaLIPA* in the *P. tsukubaensis* strain 1E5 [162]. *PtMAC2p* was found to play a crucial role in catalyzing the acylation process specifically at the C-3’ position of the mannose in MELs [163]. Knocking out acyltransferase (*PtMAC2*) or acetyltransferase (*PtMAT1*) in *P. tsukubaensis* 1E5 led to a producer of the diastereomer type of monoacylated MEL-D [163] and MEL-D (deacetylated MEL) [164]. Deletion of the putative transporter gene *PtMMF1* in the *P. tsukubaensis* strain 1E5 (ΔPtMMF1) resulted in the production of low-hydrophobicity MELs, including monoacylated MEL-B and monoacylated MEL-D [165]. More recently, the crystal structure investigation of PtMAC2p has indicated that PtMAC2p possesses a catalytic tunnel structure at the center of the molecule, in which hydrophobic amino acid residues are concentrated near His158, a critical catalytic residue, suggesting that this region is the catalytic channel, a binding site for the fatty acid side chain of MEL (acyl acceptor) and/or acyl-coenzyme A (acyl donor) [166]. In conclusion, the structure and function of the genes involved in the regulation of MEL biosynthesis have been intensively explored in the *P. tsukubaensis* strain 1E5, which is conducive to expanding the application of MELs in industry. The identification and characterization of genes involved in MEL biosynthesis in *P. tsukubaensis* have provided valuable insights into the molecular mechanisms underlying this complex biosynthetic process.

### 4.3. Genetic Basis of MEL Biosynthesis in Moesziomyces spp.

Understanding the genetic basis of MEL biosynthesis in the *Moesziomyces* (formerly *Pseudozyma*) genus, especially *M. antarticus*, *M. rugulosus*, and *M. aphidis*, is crucial for biotechnological applications, such as the engineering of MEL-overproducing strains or the development of novel MEL analogues with improved biological activities. It was found that both *M. antarcticus* and *M. aphidis* grew and produced MELs using algae bio-oils as a carbon source [167]. The MEL biosynthetic pathway in *M. antarctica* is complex and involves multiple enzymes and genetic regulators. The gene cluster responsible for MEL biosynthesis in *M. antarctica* T-34 consists of five genes, *PaEMT1*, *PaMAC1*, *PaMAC2*, *PaMAT1*, and *PaMMF1* (Figure 5), which are similar to corresponding genes in *U. maydis*, with high-level similarities of 73, 59, 52, 75, and 53%, respectively [168]. *M. antarctica* PYCC 5048^T^ and *M. aphidis* PYCC 5535^T^ are able to efficiently grow in xylan, but the direct MEL production from xylan was only detected in *M. antarctica* PYCC 5048^T^ cultures [143]; so, further exploration is required to reveal the genes associated with this unique capability. Genetic engineering techniques have been employed to investigate MEL biosynthesis in *M. antarctica*. Through the use of gene knockout and overexpression experiments, researchers have been able to gain insights into the function of specific genes and their impact on MEL production. In 2017, Saika et al. enhanced the production of MEL-B in the recombinant *P. tsukubaensis* strain 1E5 through the introduction of genes *PaLIPAp* and *PaLIPBp* derived from *M. antarctica* T-34 [161]. In a study conducted by Saika et al. in 2019 [169], it was found that knockout of the *PaEMT1* gene in the *M. antarctica* strain GB-4 resulted in a notable decrease in PaE (*M. antarctica* esterase) activity, while the addition of different surfactants, such as MEL, resulted in the restoration of PaE activity, suggesting a close relationship between MEL biosynthesis and the production of PaE in *M. antarctica*. In conclusion, the genetic basis of MEL production in *M. antarctica* represents a fascinating area of research with potential biotechnological and medical applications. Ongoing studies in this field are expected to yield further insights into the complex biosynthetic pathway leading to MEL formation and could lead to the development of improved MEL-producing strains with enhanced biological activities.

The oleaginous yeast *M. aphidis*, a fungal species belonging to Basidiomycota, has gained significant interest due to its unique biosynthetic capabilities, particularly in the production of MELs. The entire MEL biosynthesis gene cluster, comprising *EMT1*, *MAC1*, *MAC2*, *MMF1*, and *MAT1*, has been successfully identified (Figure 5). Notably, all five relevant genes exhibit significant conservation between *M. antarctica* T34 and *M. aphidis*, with nucleotide-level similarities of 89.4%, 86.8%, 91.2%, 87%, and 86.8%, respectively; these findings suggest the existence of a similar MEL pathway in *M. aphidis* to the ones in *M. antarctica* and *U. maydis* [170]. Apart from MEL, *M. aphidis* also produces a large amount of intracellular lipids. It was revealed through gas chromatography/mass spectrometry (GC/MS) that the intracellular lipids produced by strain XM01, including C16:0, C18:0, C18:1, C18:2, and C18:3, closely resemble the lipid profiles found in other oleaginous yeast strains [121], suggesting that the intracellular lipid synthesis pathway in strain XM01 is likely comparable to that of other oleaginous yeasts. Previous research on the oleaginous yeast *Yarrowia lipolytica* demonstrated a pivotal role of specific genes YALI0E32769g (DGA1) and YALI0E16797g (LRO1) in the major triacylglycerol synthesis pathway of intracellular oil synthesis [171]. The accumulation of intracellular lipids in XM01 may disrupt the carbon flow for the synthesis of MEL. As a result, further investigations should focus on the mechanisms of intracellular lipids synthesis. To enhance the conversion rate of MEL, it is proposed that knocking out some critical genes in the intracellular oil synthesis pathway could effectively impede the carbon flow toward intracellular oil production. In conclusion, the genetic basis of MEL biosynthesis in *Moesziomyces* spp. is complex and involves multiple genes encoding enzymes that catalyze distinct steps in the pathway. Understanding the functions and interactions of these genes and their encoded enzymes is essential for manipulating MEL biosynthesis and developing novel MEL analogues with improved biological activities. Future studies in this area could lead to the development of MEL-based bio-products with potential applications in medicine, agriculture, and biotechnology.

## 5. Conclusions and Perspective

MELs are a fascinating class of glycolipids with diverse biological activities that make them promising candidates for various applications in medicine, agriculture, and biotechnology [9,172,173]. Their immunomodulatory properties, antifungal activity, surfactant behavior, and potential for industrial production make MELs an exciting area of research that is likely to yield further insights and applications in the future. MELs are produced by diverse fungal strains (Table 1), and the biosynthesis of MELs is a complex process that involves multiple enzymes and intermediates (Figure 4). The availability of genome sequences is critical to elucidating the genetic basis of MEL biosynthesis, and the annotated genomes are mined for candidate genes that are potentially involved in MEL biosynthesis. Extensive clustering of functionally co-regulated gene families exists in closely related Basidiomycota fungi and allows for the stabilization of expression patterns [174]. The gene cluster encoding MELs consists of five genes: *MAT1*, *MMF1*, *MAC1*, *EMT1*, and *MAC2* [155,175] (Figure 5). As the number of *Basidiomycetes* and some other fungal lineages within a sequenced genome expand, thorough and systematic comparisons of their genomic data will illuminate insights throughout this phylum and will help to better understand the mechanisms by which these clusters form and function and ultimately to genetically manipulate these gene clusters.

It has been more than half a century since the initial discovery of MEL [18], but the direct application of MELs in many fields, such as agriculture, food, medicine, pharmaceuticals, and cosmetics, is still in its infancy. A major challenge for future industrial applications of MELs is to customize the biosynthesis of the desired MEL molecules. Genetic manipulation of the *MAC* gene in wild-type strains is expected to produce new acetylated variants of the MEL molecule. The fungal repertoire of MAC enzymes can be tailored to specific application needs using the appropriate biotechnologies, theoretically producing a large number of different MELs with predefined compositions [17]. To expand the utility of these biosurfactants, multiple studies have sought to tailor the production of targeted MELs. Genetic engineering of the MEL biosynthetic pathway will drive the creation of hosts that produce new MEL derivatives, expanding the use of MELs in industry. Although there are still many challenges in the efficient production of MELs, with the rapid development of genome sequencing and genetic manipulation technologies, as well as the increasing understanding of the MEL synthesis mechanism, we believe that the efficient production of MELs will become a reality in the near future, and targeted modification of MEL producers and tailor-made MELs will also be around the corner. This will not only provide new solutions for solving current problems in medicine, food, and cosmetics but will also promote economic development and social progress.

## Figures and Tables

**Figure 1 biomolecules-14-00557-f001:**
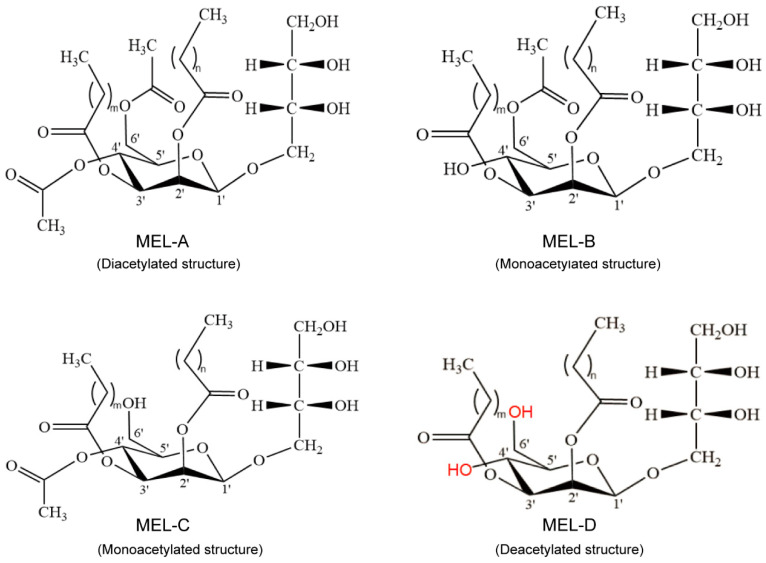
The chemical structures of four MEL homologs, MEL-A, MEL-B, MEL-C, and MEL-D. These homologs are categorized according to the carbon acetylation in the C-4 and C-6 (mannose).

**Figure 2 biomolecules-14-00557-f002:**
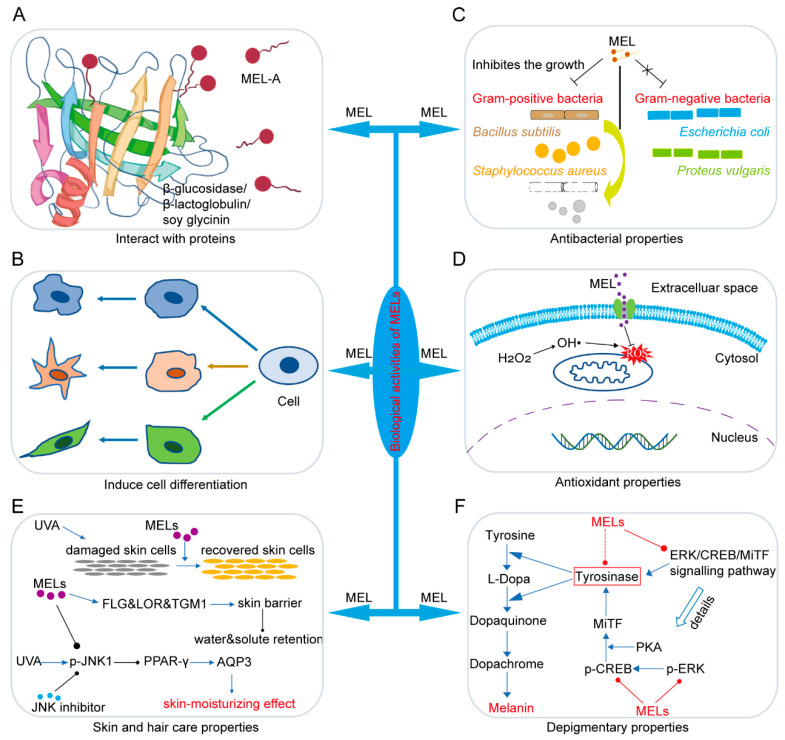
A diagram showing the various biological properties of mannosylerythritol lipids. (**A**) MELs can interact with proteins. (**B**) MELs can induce differentiation of mammalian cells. (**C**) MELs have antibacterial activity. (**D**) MELs have antioxidant activity. (**E**) MELs exhibit skin- and hair-moisturizing efficacy. (**F**) MELs have depigmentary properties.

**Figure 3 biomolecules-14-00557-f003:**
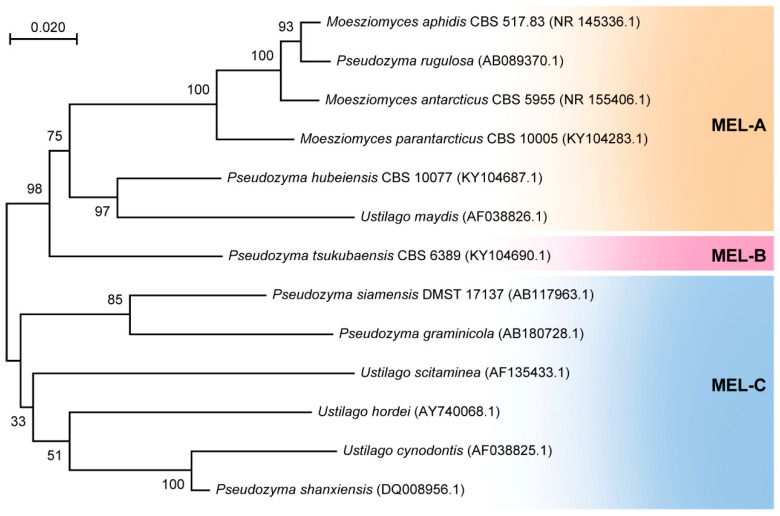
Molecular phylogenetic tree of some MEL-producing microorganisms. The phylogenetic tree was constructed using Mega X [122] based on the internal transcribed spacer (ITS). All sequences were retrieved from GenBank (https://www.ncbi.nlm.nih.gov/genbank/ accessed on 26 December 2023).

**Figure 4 biomolecules-14-00557-f004:**
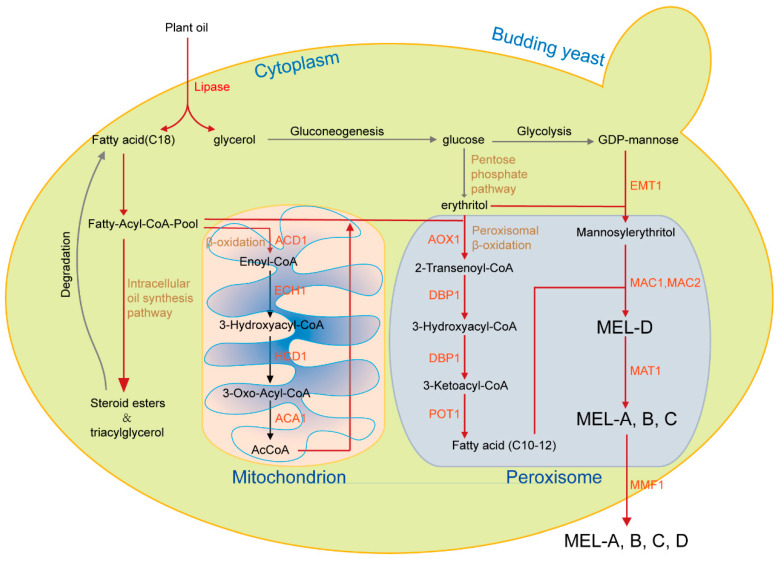
A schematic representation of the pathways and enzymes necessary for MEL biosynthesis in basidiomycetous yeasts belonging to the genera *Ustilago*, *Pseudozyma*, *Moesziomyces*, and *Sporisorium*. The genes for MEL biosynthesis encode four enzymes (EMT1, MAC1, MAC2, and MAT1) and one transport protein MMF1. EMT1 is needed for the synthesis of the hydrophilic carbohydrate backbone 4-O-β-D-mannopyranosyl-erythritol. Two peroxisomal acyltransferases Mac1 and Mac2 link MELs to fatty acid oxidation. Mat1 catalyzes the formation of acetylated MEL variants MEL-A, MEL-B, and MEL-C. The synthesis of the nonacetylated variant MEL-D does not need MAT1.

**Figure 5 biomolecules-14-00557-f005:**
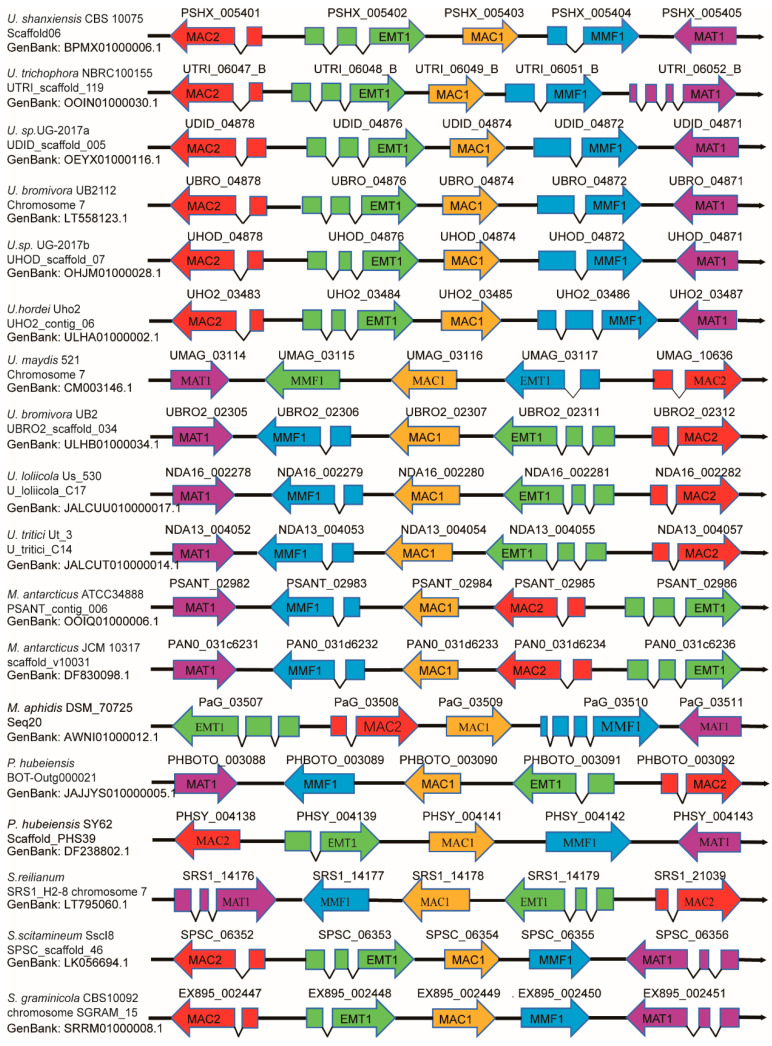
Gene clusters for the biogenesis of MEL in diverse strains. The exons of each of the five genes involved in MEL synthesis were labeled. The schematic diagram for all cluster genes is drawn according to their genomic loci shown in GenBank.

## Data Availability

No data were generated for the research described in this article.

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
