# Peer review of "Promising Application, Efficient Production, and Genetic Basis of Mannosylerythritol Lipids"

_biomolecules, 2024, doi:10.3390/biom14050557_

Round 1

Reviewer 1 Report

Comments and Suggestions for Authors

This review systematically summarizes the results of previous research on mannosylerythritol lipids, contains a large amount of information, and is thought to be very useful for biosurfactant researchers. However, there were some small mistakes, so the authors are suggested to revise the manuscript considering below-mentioned comments.

L533

three acetyltransferases => an acetyltransferase (mat1) and two acyltransferases (mac1 and mac2)

L576-577

MEL-D => mono-acylated MEL-D (ref. 165) and MEL-D (ref. 166)

L610-612

This sentence is about P. tsukubaensis, so it should be included in Chapter 4.2. Furthermore, ref. 171 is the same paper as ref. 165. I would like a correction.

L616-618

Similar to the above, this sentence is about P. tsukubaensis, so it should be included in Chapter 4.2. Furthermore, ref. 173 is the same paper as ref. 167. I would like a correction.

References

As mentioned above, there are some duplications, and some of the cited documents have author's initials (128, 129, 134), so it seems that the notation is not accurate. I would like to ask the authors to check the list of cited references again.

Fig. 2

The authors should state what the phylogenetic tree is based on (ITS or 26/28S rRNA A1/D2?).

Author Response

Author's Reply to the Review Report (Reviewer 1)

Comments and Suggestions for Authors

This review systematically summarizes the results of previous research on mannosylerythritol lipids, contains a large amount of information, and is thought to be very useful for biosurfactant researchers. However, there were some small mistakes, so the authors are suggested to revise the manuscript considering below-mentioned comments.

Responding: We are very grateful to you dear professor for reviewing our manuscript, and your valuable comments and suggestions are very important to improve our work.

L533

three acetyltransferases => an acetyltransferase (mat1) and two acyltransferases (mac1 and mac2)

Responding: Thank you dear professor. The “three acetyltransferases” were changed to “an acetyltransferase (mat1) and two acyltransferases (mac1 and mac2)” in the updated version.

L576-577

MEL-D => mono-acylated MEL-D (ref. 165) and MEL-D (ref. 166)

Responding: Thank you dear professor. “MEL-D” was changed to “mono-acylated MEL-D [165] and MEL-D (deacetylated MEL) [166]”.

L610-612

This sentence is about P. tsukubaensis, so it should be included in Chapter 4.2. Furthermore, ref. 171 is the same paper as ref. 165. I would like a correction.

Responding: Thank you dear professor. This sentence has been included in Chapter 4.2. The ref. 171 was replaced by ref. 165.

L616-618

Similar to the above, this sentence is about P. tsukubaensis, so it should be included in Chapter 4.2. Furthermore, ref. 173 is the same paper as ref. 167. I would like a correction.

Responding: Thank you dear professor. This sentence was deleted. The ref. 173 was replaced by ref. 167.

References

As mentioned above, there are some duplications, and some of the cited documents have author's initials (128, 129, 134), so it seems that the notation is not accurate. I would like to ask the authors to check the list of cited references again.

Responding: Thank you very much. We have checked all references one by one to ensure every reference and its citing are correct.

Fig. 2

The authors should state what the phylogenetic tree is based on (ITS or 26/28S rRNA A1/D2?).

Responding: Thank you very much. The phylogenetic tree is based on the internal transcribed spacer region (ITS) of the rRNA gene cluster, which was stated in the figure legend. All sequences were retrieved from GenBank.

Reviewer 2 Report

Comments and Suggestions for Authors

Re: Promising application, efficient production and genetic basis of mannosylerythritol lipids, submitted to Biomolecules , by Liu, Liu and Liu.

This review is well written and thorough, covering the important aspects of MEL applications, means of production, and biosynthesis, with emphasis on likely ways in which to improve yields, simplify purification and isolate desired subfractions.  The figures are of distinctly high quality.  As such, I highly recommend publication, subject only to the consideration mentioned below.

I had to wait until page 8, fig 2 before finally encountering a chemical structure for MEL-A or -B or -C or -D, i.e. nearly 1/3 of the way through the review.  In my opinion, the chemical structures should be featured early on, for instance in the first paragraph of the introduction where they are described in words.  These words should be accompanied by a separate figure explicitly showing their chemical structures.  (It would then not be necessary to include their structures in Fig 2 on page 8.)

Some minor suggestions:

Page 2 line 70 “influences” rather than “accounts for” might be more appropriate.

Comments on the Quality of English Language

English is fine.

Author Response

Comments and Suggestions for Authors

Re: Promising application, efficient production and genetic basis of mannosylerythritol lipids, submitted to Biomolecules, by Liu, Liu and Liu.

This review is well written and thorough, covering the important aspects of MEL applications, means of production, and biosynthesis, with emphasis on likely ways in which to improve yields, simplify purification and isolate desired subfractions. The figures are of distinctly high quality. As such, I highly recommend publication, subject only to the consideration mentioned below.

Responding: Thank you very much for your valuable comments and suggestions which are very important for us to improve this manuscript.

I had to wait until page 8, fig 2 before finally encountering a chemical structure for MEL-A or -B or -C or -D, i.e. nearly 1/3 of the way through the review. In my opinion, the chemical structures should be featured early on, for instance in the first paragraph of the introduction where they are described in words. These words should be accompanied by a separate figure explicitly showing their chemical structures. (It would then not be necessary to include their structures in Fig 2 on page 8.)

Responding: Thank you very much. We have added the figure (Figure 1) in the first paragraph, and changed the numbering of the other figures and their citing throughout the text accordingly.

Some minor suggestions:

Page 2 line 70 “influences” rather than “accounts for” might be more appropriate.

Responding: Thank you very much. The word “influences” here was replaced by “accounts for”.

Round 2

Reviewer 1 Report

Comments and Suggestions for Authors

Accept